# Refining Stereotaxic Neurosurgery Techniques and Welfare Assessment for Long-Term Intracerebroventricular Device Implantation in Rodents

**DOI:** 10.3390/ani13162627

**Published:** 2023-08-14

**Authors:** Ester Pérez-Martín, Almudena Coto-Vilcapoma, Juan Castilla-Silgado, María Rodríguez-Cañón, Catuxa Prado, Gabriel Álvarez, Marco Antonio Álvarez-Vega, Benjamín Fernández-García, Manuel Menéndez-González, Cristina Tomás-Zapico

**Affiliations:** 1Neuroscience Innovative Technologies S.L., Neurostech, 33428 Llanera, Spaincatuxa.prado@neurostech.com (C.P.);; 2Departamento de Biología Funcional, Área de Fisiología, Universidad de Oviedo, 33006 Oviedo, Spain; 3Instituto de Investigación Sanitaria del Principado de Asturias (ISPA), 33011 Oviedo, Spain; 4Departamento de Cirugía, Área de Cirugía, Universidad de Oviedo, 33006 Oviedo, Spain; 5Servicio de Neurocirugía, Hospital Universitario Central de Asturias, 33011 Oviedo, Spain; 6Departamento de Morfología y Biología Celular, Área de Anatomía, Universidad de Oviedo, 33006 Oviedo, Spain; 7Servicio de Neurología, Hospital Universitario Central de Asturias, 33011 Oviedo, Spain; 8Departamento de Medicina, Universidad de Oviedo, 33011 Oviedo, Spain

**Keywords:** 3Rs, animal research, animal welfare, cannula fixation, intrathecal implantation, preclinical neurosurgery, reduction, refinement, stereotaxic surgery

## Abstract

**Simple Summary:**

The development of innovative therapeutic strategies involving chronic drug delivery to specific brain regions has been crucial in preclinical neuroscience. However, these strategies involve complex surgeries due to the need for implanting a drug storage system connected to the brain through a cannula while ensuring animal welfare, which is a challenge, especially for long-term studies in rodents. In this study, we propose an optimized method with three main refinements: (i) modifying the dimensions of the implantable devices, (ii) using a combination of adhesive tissue and UV light-curing resin, and (iii) implementing a customized scoresheet to closely monitor animal welfare throughout the experiment. Overall, the proposed refinements significantly improved animal welfare, reduced complications related to surgery, increased animal survival, and ensured safe long-term implantations.

**Abstract:**

Stereotaxic surgeries enable precise access to specific brain regions, being of particular interest for chronic intracerebroventricular drug delivery. However, the challenge of long-term studies at this level is to allow the implantation of drug storage devices and their correct intrathecal connection while guaranteeing animal welfare during the entire study period. In this study, we propose an optimized method for safe intrathecal device implantation, focusing on preoperative, intraoperative, and postoperative procedures, following the 3Rs principle and animal welfare regulations. Our optimized protocol introduces three main refinements. Firstly, we modify the dimensions of the implantable devices, notably diminishing the device-to-mouse weight ratio. Secondly, we use a combination of cyanoacrylate tissue adhesive and UV light-curing resin, which decreases surgery time, improves healing, and notably minimizes cannula detachment or adverse effects. Thirdly, we develop a customized welfare assessment scoresheet to accurately monitor animal well-being during long-term implantations. Taken together, these refinements positively impacted animal welfare by minimizing the negative effects on body weight, surgery-related complications, and anxiety-like behaviors. Overall, the proposed refinements have the potential to reduce animal use, enhance experimental data quality, and improve reproducibility. Additionally, these improvements can be extended to other neurosurgical techniques, thereby advancing neuroscience research, and benefiting the scientific community.

## 1. Introduction

Stereotaxic surgeries, in combination with other cutting-edge approaches, have revolutionized preclinical neuroscience research [1]. These techniques have enabled precise and controlled access to specific regions of the nervous system, offering unprecedented opportunities to study brain function and evaluate potential therapeutic approaches, including chronic intracerebral or intraventricular administration of drugs, in vivo optogenetic manipulations for activating or inactivating brain structures or systems, and in vivo electrophysiological recordings in awake behaving rodents (Refs. [1,2,3,4,5,6,7], among others). However, the success of these techniques relies heavily on complex, highly precise, and time-consuming surgical procedures, the failure of which not only entails a significant loss in economic and human resources, but also compromises the welfare of the animals involved [8]. In fact, we have recently conducted a proof-of-concept study to assess the feasibility and safety of implanting a device for continuous cerebrospinal fluid (CSF) apheresis [2]. In this study, more than 30% of the operated mice had to be euthanized following humane criteria. Retrospective analysis of the causes of this high mortality rate led us to identify two factors that may have the largest effect at this level: (i) the size and weight of the device, which accounted for more than 65% and 10% of the operated mice’s size and body weight, respectively [2]; and (ii) the correct fixation of the cannula, allowing freedom of movement of the animals without detachment from the skull. Based on our experience, this last aspect stands out as the most critical one since it entails the formation of wounds that do not heal and undergo necrosis, becoming the main reason for euthanasia of the affected animals.

Although adjusting the size of implantable devices would be one of the simplest refinement procedures, this does not apply to cannula fixation. In fact, one of the main challenges and limiting factors of these procedures in small rodents is the secure fixation of cannulas, guides, or electrodes at the specific stereotaxic coordinates on the skull surface for long-term studies. Traditionally, three methods have been employed: (i) small anchoring screws combined with dental cement (zinc-polycarboxylate), (ii) a combination of white and pink dental cement (zinc-polycarboxylate and methyl-methacrylate, respectively), or (iii) cyanoacrylate adhesive gel [1,9,10]. Despite the benefits of the aforementioned options, all of them pose drawbacks not only for experimental rodents, but also for researchers as some of these compounds contain respiratory tract irritants [11]. Regarding the reported complications in small rodents, the use of dental cements or cyanoacrylate has been associated with an increased incidence of surgical problems, such as skin necrosis, brain damage and trauma, infection, and, most frequently, cannula detachment due to the round shape of mouse skull. To address the last limitation, Sike et al. proposed in 2017 an improved method involving the development of a custom skull-shaped silicone spacer as a fixation adapter to be used in combination with cyanoacrylate tissue adhesive [10]. This method reduced the time of surgery compared to dental cement and minimized adverse effects, although it significantly increased the time required for preoperative procedures. In addition, it required specific materials and equipment, such as a 3D printer or microCT scanner, as the silicone spacer was reconstructed and molded from an original mouse skull, making it a non-universal method and not affordable for all researchers.

Following the European Directive 2010/63/EU (for review [12]) for the protection of laboratory animals and adhering to the 3Rs principle (replacement, reduction, and refinement) from *The Principles of Humane Experimental Technique* [13], in this work, we propose a new refined method to securely performed stereotaxic surgeries involving intracerebroventricular or intrathecal device implantation in long-term studies. These refinements focused on the main critical steps of intrathecal implantation in rodents (i.e., preoperative, intraoperative, and postoperative cares and procedures [14]) and were based on our experience in different studies conducted in our laboratory. Our proposed protocol introduces three significant changes. Firstly, we have modified the dimensions of the implantable devices we have previously used [2], while ensuring the functionality of the device. This allowed us to significantly reduce the ratio between the weight of the devices and the body weight of the mice. Secondly, we use a combination of cyanoacrylate tissue adhesive and UV light-curing resin, which significantly reduces surgery time, improves wound healing, and reduces the postoperative recovery period, with a near 100% success rate. Thirdly, we have designed a customized welfare assessment scoresheet to monitor animals undergoing long-term cannula implantation, including indicators that accurately and effectively reflect animals’ well-being for this particular surgery [15,16,17,18,19,20,21,22,23]. By implementing all the improvements, we evidenced a positive impact on animal welfare, the reduction in the number of animals used, and the quality of the experimental data, all of which align with the “refinement” and “reduction” principles of the 3Rs [13]. Furthermore, these advances hold the potential to be applied in other neurosurgical techniques that require long-term implantation, including optic fiber fixation for optogenetics or electrode placement for in vivo electrophysiological recordings, among others, which may greatly benefit the neuroscience community and research.

## 2. Materials and Methods

### 2.1. Animals

Seven/eight-month-old male transgenic APP/PS1 (*n* = 40, hereafter referred to as “APP”) of the 129Sv strain background mice expressing the chimeric mouse/human amyloid precursor protein (Mo/HuAPP695swe) and the mutant human presenilin 1 (PS1-dE9; [2,24]) and their non-transgenic wild-type (WT) littermates (*n* = 32) were used in this study. The animals were randomly assigned to one of four experimental groups depending on the intrathecal device implanted and the neurosurgery protocol followed (Figure 1a): (i) naïve (*n* = 10 for each genotype); (ii) group implanted with the original device using traditional surgical techniques (*n* WT = 9; *n* APP = 10) from [2]; (iii) group implanted with the miniaturized device using optimized surgical techniques (*n* WT = 3; *n* APP = 10); and (iv) group implanted with a commercial osmotic pump using optimized surgical techniques (*n* = 10 for each genotype; see specifications of the different implantable devices in Section 2.2).

Considering that the experiment end times varied among the different experimental groups, which are part of larger studies, our primary focus in this work was to analyze the parameters at timepoints shared by most of the groups, mainly week (W)-1, W3, and W8. The exact sample sizes for each parameter and timepoint analyzed are provided in Table 1. In addition, the experimental groups, design, and timeline are depicted in Figure 1a and will be described in detail in the following sections.

The mice were housed under a 12/12 light/dark cycle (with lights on at 8:00 a.m.) at both constant room temperature (22 ± 2 °C) and relative humidity (55 ± 7%) and provided with ad libitum access to water and rodent global diet chow (A40; SAFE^®^, Rosenberg, Germany) at the animal facilities of Universidad de Oviedo (Spain). All animal procedures were performed during the light period (8:30–15:00 a.m.) and were approved by the Research Ethics Committee of the University of Oviedo (PROAE IDs 32/2020; 05/2022 and 06/2022) in compliance with European (Directive 2010/63/UE) and Spanish (RD118/2021, Law 32/2007) legislation. Every possible effort was made to ensure animal welfare, minimize pain and distress, and employ the smallest sample size necessary to achieve statistically relevant results, in accordance with the ARRIVE guidelines developed by the NC3R [25].

### 2.2. Implantable Devices

In our study, we employed different intrathecal implantable devices, namely, the original device [2], miniaturized device, and commercial pump (Figure 1b, Table 2). Both the original and miniaturized devices correspond to two prototypes of a medical device designed and manufactured by Neuroscience Innovative Technologies S.L. (https://neurostech.com/; accessed on 31 May 2023) for continuous and selective apheresis of CSF (in-depth review of this innovative therapeutic strategy [26,27,28]). Briefly, these devices comprise two main components: a subcutaneous reservoir and an apheresis module with selective nanoporous membranes connected to an intrathecal brain infusion cannula (Brain Infusion Kit 3; ALZET^®^, Cupertino, CA, USA). Their unique design allows the clearing of toxic molecules from CSF while avoiding the side effects caused by the direct contact between the therapeutic agent and the brain parenchyma [2,27,28]. The main and pivotal distinctions between the original and miniaturized devices lie in the dimensions of the device components and the change in the coating material, as outlined in Table 2 and Table 3, without compromising its functionality. These modifications result in the following benefits: (i) a reduction in total weight and volume by 47.55% and 57.09%, respectively; (ii) a decrease in length, width, and height by 17.44%, 13.33%, and 40%, respectively; and (iii) a drop in the device weight-to-animal body weight ratio from 14.16 ± 1.35% to 6.93 ± 0.46% (equating to a 51.05% reduction). Additionally, we included the ALZET^®^ 1004 osmotic pump (referred to as “Alzet pump” hereafter; Figure 1b) as a commercial implantable device with smaller dimensions than the miniaturized device for comparative purposes (Table 2 and Table 3).

Preparation of the implantable devices was conducted prior to surgery. To begin with, the original and miniaturized devices were sterilized by exposure to 25 kGy gamma irradiation from a ^60^Co source, following a similar sterilization process as that specified in ALZET^®^ products and other medical instruments. On the day of surgery, the filling and priming of implantable devices were performed under sterile conditions as previously described [2] and adhering to the manufacturer’s instructions. The original and miniaturized devices were loaded with a solution containing non-conjugated anti-β-Amyloid antibody or artificial CSF (aCSF; Tocris Bioscience, Bristol, UK). Alzet pumps were filled with a recombinant mouse factor or aCSF.

Since no significant differences in body weight and behavior were observed between animals receiving the therapeutic agent and those receiving the vehicle at the analyzed timepoints employed in this study, all animals were included in the same experimental group regardless of the treatment.

### 2.3. Intrathecal Implantable Device Surgery

The following sections will describe the optimized protocol for intrathecal device implantation surgery, with a particular emphasis on the refined steps that have been proven to be essential in ensuring and enhancing the welfare of mice during long-term implantations compared to existing protocols [2,4,5,7,8,10].

#### 2.3.1. Preoperative Care and Treatment

Regardless of the implanted device and surgical protocol followed, all animals received a presurgical analgesic and antibiotic cocktail 15–20 min prior to the start of the surgery. This treatment consisted of buprenorphine (0.05 mg/kg, subcutaneous injection (SC), Bupaq; Richter Pharma, Wels, Austria), enrofloxacin (10 mg/kg, SC, Syvaquinol; Syva Laboratories, Leon, Spain). Moreover, sterile saline solution was administered intraperitoneally (IP) or SC (1 mL of 0.9% (*w*/*v*) NaCl; B. Braun, Melsungen, Germany) to prevent dehydration associated with the surgical procedure itself.

To minimize the high risk of aggression and potential wound opening/infection in male mice sharing housing, all animals were individually housed. In the case of animals implanted with the original device, immediate isolation following surgery was carried out, which may have contributed to increased distress. To address this concern, in the refined protocol (i.e., animals implanted with the miniaturized device and Alzet pump) individual isolation was conducted one week prior to surgery (W1). Additionally, a diverse range of cage enrichments, such as a red polycarbonate arch (Bio-Serv, Flemington, NJ, USA), sizzle-nest (Datesand, Bredbury, UK), disposable cardboard tunnels, cocoon (Datesand), or paper wool (Datesand) were simultaneously provided. Moreover, palatable gel diet energy and transport food (SAFE^®,^, Rosenberg, Germany) in a Petri dish was placed in the cage bedding. This approach allowed the animals to adapt to their new housing conditions for a 7-day acclimation period prior to surgery, minimizing possible stress and anxiety associated with novel food and environments [31,32].

On the day of surgery, animals implanted with miniaturized and Alzet pumps were placed inside a veterinary intensive care unit (S50 Advance Series II Model; Vetario, Westen-Super-Mare, UK) and maintained at a controlled temperature of 33–36 °C for 30 min, as pre-anesthetic warming has been shown to prevent hypothermia-related complications and to shorten recovery from anesthesia in rodents [33,34,35].

#### 2.3.2. Stereotaxic Surgery Procedure

Depending on the experimental group, the protocol employed for implantation of the intrathecal device varied. For animals implanted with the original device, traditional protocols were followed (as detailed in [2]). However, for animals implanted with the miniaturized device or Alzet pump, a refined technique was used. In this section, we will primarily focus on the modifications made to the stereotaxic neurosurgery protocol for long-term intrathecal device implantation compared to the available traditional protocols [2,4,5,7,8]. A detailed step-by-step protocol is provided in Appendix A.

A crucial aspect of all surgeries is the meticulous preparation of the operating room and surgical instruments, which involves maintaining “clean” and “dirty” areas clearly separated (see Figure 2). Every effort was made to employ strict aseptic techniques and to ensure a sterile surgical field throughout the entire surgical process. For instance, the working space was first cleaned with soap and water, dried, and then wiped with 70% ethanol. The surface was covered with protective paper (Nalgene™ Super Versi-Dry™ Surface Protectors, Thermo Scientific™, Waltman, MA, USA), allowing the sterilized material to be placed on it, organized as shown in Figure 2. Prior to each procedure and between surgeries on different animals, all instruments underwent a thorough cleaning with an enzymatic detergent (Helizyme; B. Braun), followed by sterilization both with 70% (*v*/*v*) ethanol and by heating at 230 °C for 1 min in a microbead sterilizer (B1305-E-FIS; Fisherbrand, Waltman, MA, USA). In addition, all the researchers involved in the surgery and in assisting the surgery changed gloves between mice and avoided touching other areas of the operating room that may have been dirty. On the other hand, the larger equipment (isoflurane delivery system, surface of the induction chamber, the stereotaxic frame and its digital controller, and the stereo microscope), as well as the containers that carried the medication or products to be used during surgery, were cleaned with 70% ethanol. All the surgeries were performed from 08:30 to 15:00 by the same surgeons (E.P.-M. and M.A.Á.-V.) and surgery assistants (A.C.-V.; J.C.-S. and B.F.-G.).

After pre-anesthetic warming the animals and administering the presurgical analgesic and antibiotic cocktail (see details in the previous section), anesthesia was induced with 4% (*v*/*v*) isoflurane (IsoFlo; Zoetis, Parsippany, NJ, USA) and 2.5% (*v*/*v*) O_2_. The animals were gently shaved on top of the head (Aesculap Isis Rodent Shaver; Aesculap Schermaschinen, Buschbach, Germany) and a topical anesthetic ointment (lidocaine 25 mg/g and prilocaine 25 mg/g, EMLA; AstraZeneca, Cambridge, UK) was applied around the ear canal (Figure 3). Thereafter, the animal was securely placed in a digital stereotaxic frame (Harvard Apparatus, Holliston, MA, USA) with a heating pad underneath to prevent hypothermia during the procedure. Temperature monitoring using a rectal thermometer connected to a physiological monitoring system (Harvard Apparatus) is strongly recommended. Once the absence of withdrawal reflex was confirmed and ocular ointment (Lubrithal; Dechra Pharmaceuticals, Northwich, UK) was applied, a U-shaped incision was made in the specific area between the two auditory pavilions, and the upper skin was removed rostrally by using two hemostats (see Figure 3). To facilitate visualization of the craniometric points bregma and lambda on the cranial surface and to enhance posterior cannula fixation, the periosteum was scraped using a microcurette (Model 10080-05; Fine Surgical Tools F.S.T., Chandler, AZ, USA) and dried with a cotton swab moistened with 70% (*v*/*v*) ethanol. The coordinates of the bregma and lambda landmarks were then measured to accurately position them in the dorso-ventral (DV), medio-lateral (ML) and antero-posterior (AP) planes [1,36,37] and readjusted whenever the difference between coordinates exceeded 0.05 mm. Once a “flat-skull” position was reached, a 0.5 mm diameter craniotomy was drilled in the skull at the specific coordinates AP= −0.70 mm and ML= −1.26 mm (Figure 3). Subsequently, a subcutaneous pocket was created with a hemostat by dissecting from the original U-incision towards the mouse’s flanks, and the implantable device was carefully inserted. The intraventricular cannula was slowly inserted into the craniotomy until DV= −2.5 mm using a cannula holder (Stoelting Co., Wood Dale, IL, USA) and was glued to the skull surface with cyanoacrylate adhesive tissue (Cicastick; Sutuvet, Santiago de Surco, Peru). To secure the cannula in place, a thin annular layer of UV light-curing resin (Transbond XT 4; 3M, Saint Paul, MN, USA) was applied around the base of the brain cannula and polymerized in 3 cycles of 20 s using a portable UV lamp (Lux E; Bestdent, Gansu, China). After the resin was fully hardened (1 min), the upper tip of the cannula was removed using a bone nipper (Model 16102-11, F.S.T.; Figure 3). The previously removed skin was rehydrated with sterile saline solution and the incision was closed with single stitches using 5-0 monofilament non-resorbable polypropylene suture (Atramat^®^, Mexico City, Mexico). It was crucial to increase the density of stitches at the position of maximum tension of the U-shaped suture, corresponding to the portion where the catheter was positioned beneath the skin (Figure 3). Finally, a 1% (*v*/*v*) povidone-iodine solution (Betadine; Mylan, Canonsburg, PA, USA) and topical antibiotic ointment (Fucidine 20 mg/g; LEO Pharma, Ballerup, Denmark) were applied using a sterile cotton swab. The isoflurane anesthetic was gradually reduced, and the animal was carefully removed from the stereotaxic frame. Recovery time from anesthesia was recorded and classified into the following intervals: >20 min, 5–10 min, or <5 min.

#### 2.3.3. Postoperative Care

Immediately after surgery, while the animals were still recovering from anesthesia, the nails of the front and rear feet were carefully trimmed to prevent scratching that could worsen incision healing or lead to wound reopening (Figure 3). It is important to note that only the most distal portion of the nail plate, identifiable by its “pearly” coloration, should be trimmed to avoid bleeding. Subsequently, anti-inflammatory treatment was administered using meloxicam (2 mg/kg, SC, Metacam; Boehringer Ingelheim, Rhein, Germany), and rehydration was ensured by supplying sterile saline solution (1 mL, SC). If necessary, sterile saline with 5% (*w*/*v*) glucose (B. Braun) can be SC administered. The mice were then returned to a clean and enriched cage (Figure 3) located within the veterinary intensive care unit, and received a postoperative regimen of analgesic, anti-inflammatory, and antibiotic treatment for two weeks following surgery [38], as described in Appendix A.

### 2.4. Body Weight Monitoring and Welfare Assessment

Body weight was measured weekly between 09:00 and 13:00 p.m., starting one week before surgery (referred to as “basal body weight” or W1) and continuing until the end of the experiment, using a portable balance (CB 501; Adam Equipment, Milton Keynes, UK), as the body weight reduction is widely used as a humane endpoint (Refs. [39,40], among others). From the week of surgical implantation onwards, the actual body weight of the animals was inferred by subtracting the weight of the implanted device from the total body weight measured. Then, the percentage change in body weight for subsequent weeks was calculated as follows:(1)$$\%\;\text{Change in Body weight}=\left(\frac{\text{Basal W}-1\;\text{Body weight}-\text{Real Body weight}}{\text{Basal W}-1\;\text{Body weight}}\right)\times 100$$

Regarding welfare evaluation, all implanted animals were closely and intensively monitored for 24 h, which is considered the critical period. For the remainder of the experiment, daily monitoring and welfare assessment were performed every morning (9:00–13:00 p.m.). All observations were made and diligently documented on a shared data sheet by blinded scorers/observers (E.P.-M. and A.C.-V. or J.C.-S.; an example of the “postoperative monitoring form” is provided in Appendix A). To ensure consistency and reproducibility, a scoresheet was designed, including several indicators related to general condition, nutritional and hydration status, spontaneous behavior, and surgery-related parameters, as well as compensatory measures, as detailed in Table 4 [15,17,20,21,22,39,40]. Overall, these welfare indicators were carefully chosen based on our experience with these specific surgical procedures and their relevance, ease of recognition, reliability, and effectiveness in providing accurate assessment of welfare [17]. 

As indicated in Table 4, if compensatory measures failed to alleviate complications or welfare scores reached values above 12 points, a humane endpoint was implemented, leading to the euthanasia of animals. Complications and the percentage of animals reaching the humane endpoint were recorded for each experimental group (see details in Appendix A).

### 2.5. Behavioral Analysis

To explore general and anxiety-like behaviors [41], the elevated zero maze was performed both before (W1) and after surgery (W4/5 and W8, according to experimental groups; for specific sample size see Table 1) between 09:00 and 11:00 am. The apparatus (UGO Basile, Gemonio, Italy) consists of an annular gray platform 60 cm in diameter, positioned 60 cm above the floor. It features two closed corridors of 15 cm in length. For the test, a single mouse was placed in the center of one of the closed areas, and 5 min video recordings were captured using a Basler ace acA1300-60gm GigE camera (Basler, Ahrensburg, Germany). The acquired videos were automatically analyzed using the Ethovision XT version 16 software for Windows (Noldus, Wageningen, The Netherlands).

The rodents’ instinctive tendency to explore novel environments and to avoid unprotected open and elevated spaces were analyzed based on three different measures: total distance traveled, and the percentage of time spent exploring the open areas (TO) or closed areas (TC), calculated as follows:(2)$$\%\;\text{Open areas}=\frac{T_{O}}{T_{O}+T_{C}},\;\%\;\text{Closed areas}=\frac{T_{C}}{T_{O}+T_{C}}$$

Following each trial, the elevated zero maze apparatus was cleaned with a 70% (*v*/*v*) ethanol solution to avoid the influence of odors.

To ensure minimal impact on the overall animal well-being, behavioral testing was exclusively conducted in the following experimental groups: naïve, animals implanted with the miniaturized device, and animals implanted with Alzet pumps, as these animals showed higher welfare assessment scores (see Section 3). This approach aimed to minimize any potential additional adverse effect or stress on animals, which could potentially worsen their welfare.

### 2.6. Catheter Placement Validation

After the experiments, correct catheter placement at the specific stereotaxic coordinates in the right lateral ventricle was confirmed by two different methods detailed below.

#### 2.6.1. In Vivo Dye Infusion and Visualization

Firstly, a solution of 5% (*w*/*v*) blue dextran (5 kDa; TdB Labs, Ultuna, Arlanda) dye diluted in aCSF was infused by percutaneous access through the reservoir of one of the animals implanted with the miniaturized device. A total of 200 µL of tracer solution was infused in two 24 h interval sessions, at a flow rate of 5 µL/min, using a programmable syringe pump (Remote Infuse/Withdraw Pump 11 Elite Nanomite Programmable Syringe Pump; Harvard Apparatus). Following CSF tracer infusion, while the mouse was under anesthesia with 1.5–2% (*v*/*v*) isoflurane and 2.5% (*v*/*v*) O_2_, the cisterna magna was exposed using a customized protocol based on a previous study [42]. A stereomicroscope with integrated camera (S9i; Leica Microsystems, Wetzlar, Germany) was employed to visualize in vivo the CSF color in the mouse ventricular system. Subsequently, the mouse was euthanized by decapitation, and the brain immediately removed and embedded in Tissue-Tek^®^ O.C.T. (Sakura Finetek Europe, Alphen aan den Rijn, The Netherlands) before being frozen at −80 °C. The brain was then sliced into 80 µm thick transversal sections using a cryostat (CM1900; Leica Microsystems). Images were captured using a digital camera to observe the visualization of the blue/dark color tracer within the mouse ventricular system.

#### 2.6.2. Histological Analysis

To verify the specific AP, ML, and DV stereotaxic coordinates at which the intracerebroventricular catheter was placed, histological analysis was also performed. Briefly, an animal implanted with the miniaturized device was deeply anesthetized with 4% (*v*/*v*) isoflurane, and both the device and intracerebroventricular cannula were carefully removed. After exsanguination, the mouse was perfused with 40 mL of sterile ice-cold phosphate buffer saline (PBS, pH 7.4; Corning Incorporated, Corning, NY, USA) to clean tissues. The brain was immediately extracted and fixed in 4% (*v*/*v*) paraformaldehyde solution (Electron Microscopy Science, Hatfield, PA, USA) diluted in Sorensen’s phosphate buffer overnight at 4 °C with continuous shaking. After cryoprotection in a solution of 30% (*w*/*v*) sucrose (Panreac AppliChem, Darmstadt, Germany) in PBS for 24 h, the brain was frozen in Tissue-Tek^®^ O.C.T. and stored at −80 °C until further use. The brain was sectioned into 30 µm thick coronal slices using a cryostat (CM1900; Leica Microsystems), then washed in PBS and cryoprotected in a solution containing 30% (*v*/*v*) glycerol (VWR Chemicals, Radnor, PA, USA) and 30% (*v*/*v*) ethylene glycol (Sigma-Aldrich, St. Louis, MO, USA) in 0.02 M phosphate buffer, pH 7.2 at −20 °C until used for histological staining. Three non-consecutive coronal sections were washed in PBS (3 × 10 min) and stained with 0.5% (*w*/*v*) Toluidine blue O (Sigma-Aldrich) in distilled water for 1 min at room temperature. Next, the slices were washed in PBS (3 × 10 min), dried overnight at room temperature, and mounted on gelatin-coated slides (VWR). Finally, the sections were covered with an aqueous mounting medium (Aquatex^®^, Merck, Rhaway, NJ, USA). Visualization and image acquisition were performed using the integrated camera of a stereomicroscope (S9i; Leica Microsystems) and compared with Paxinos and Franklin’s Mouse Atlas [37] and the Allen Brain Mouse Atlas (http://mouse.brain-map.org/; accessed on 1 July 2023) for reference.

### 2.7. Statistics

The data are presented as individual data points and as the median ± interquartile range (IQR), unless otherwise specified. The normality of the data was assessed before conducting statistical analyses using both the Kolmogorov–Smirnov and the Shapiro–Wilk tests. Due to the non-normal distribution of most variables, as indicated by *p*-values < 0.05 in the normality tests, and the unequal sample size of the experimental groups (Table 1), the following non-parametric tests were employed in this work. For independent or dependent quantitative two-samples comparisons, the Mann–Whitney U test or the Wilcoxon test was carried out, respectively. For independent or repeated-measures quantitative multiple comparisons, the Kruskal–Wallis test or the Friedman test were, respectively, employed, followed by Dunn’s post hoc test when required. Finally, for qualitative analyses (i.e., “range of recovery time after anesthesia” and “humane endpoint”), variables were properly dichotomized when necessary and the Fisher exact test was performed. A *p*-value < 0.05 was set as the minimum level of statistical significance. All statistical analyses were conducted with SPSS Statistics version 26 for Windows (IBM, Armonk, NY, USA), and graphical representations were generated with GraphPad Prism version 9.0.2 for Windows (GraphPad Dotmatics, San Diego, CA, USA).

## 3. Results

### 3.1. The Optimized Protocol of Stereotaxic Surgeries Shortens the Recovery Time Required after Surgery and Significantly Minimizes Humane Endpoint Application

Recovery time was measured as the time interval between the end of stereotaxic surgery (i.e., withdrawal of inhaled anesthesia) and complete recovery of the animal (i.e., interaction with food or enrichment objects) and was classified in three different ranges: higher than 20 min, between 5 and 10 min, and less than 5 min, based on the data obtained (Table 5 and Figure 4a). Regarding the original device and traditional protocols, 100% (*n* = 20 out of 20) of the animals required more than 20 min to fully recover from anesthesia (Figure 4a,b). For animals implanted with the miniaturized device and following the optimized protocol, 30.76% (*n* = 3 out of 13) and 69.24% (*n* = 10 out of 13) of mice needed between 5 and 10 and 5 min to recover, respectively (Figure 4a,b). Finally, 100% (*n* = 20 out of 20) of the Alzet-pump-implanted mice following the optimized protocol recovered in less than 5 min (Figure 4a,b). These results indicate a positive impact of both the changes in the device dimensions and the implementation of a refined protocol in reducing the time required to fully recover from stereotaxic surgery (Figure 4a,b). Furthermore, the Fisher exact test revealed a clear relationship between the implanted device and recovery time range for most dichotomized variables and comparisons carried out (specific *p*-values are provided in Table 5).

Regarding the application of humane endpoint based on welfare assessment scores, changes in the device dimensions and the implementation of refined steps in the protocol strongly decreased the percentage of application, this value being 53% for mice implanted with the original device, 8% for animals implanted with the miniaturized device, and 5% for Alzet-implanted mice (Figure 4c). In addition, the Fisher exact test demonstrated a relationship between the device implanted in the experimental group and the percentage of humane endpoint application, particularly in the naïve group and in those implanted with the original device (specific *p*-values in Table 5; Figure 4c).

### 3.2. Improved Protocol Reduces the Drop in Body Weight Observed in Mice from the First Postoperative Surgery Onwards and Normalizes It over Time

Body weight monitoring was performed throughout the experiment, starting at W1 and lasting for the remainder (Figure 5), as a broadly general humane endpoint ([39,40], among others). Overall, one week before surgery (W1), a higher body weight was evidenced in APP mice compared to their WT counterparts in the naïve group and in animals that were to be implanted with the original device and Alzet pumps (*p* naïve= 0.007; *p* original = 0.043; *p* Alzet pump = 0.004; Figure 5a,b and Appendix A). Moreover, we also detected differences among animals planned to be implanted with different devices, with body weight being higher in those mice that would receive the original device compared to the rest of the mice (*p* WT naïve vs. original = 0.043; *p* WT original vs. Alzet = 0.001; *p* APP naïve vs. Alzet = 0.043; *p* APP original vs. miniaturized = 0.028; *p* APP original vs. Alzet = 0.002; Figure 5b). It is worth mentioning that the original device weighed 4.29 ± 0.41 g (see Table 2 for specifications). Therefore, mice with higher body weight values were selected to be implanted with this device in order to ensure their welfare and the feasibility of the surgery. In addition, the reduced dimensions of the miniaturized device and Alzet pump enabled us to perform the implantation surgery on animals of a lower weight (Figure 5b), which would not have been possible with the original one and explains the initial differences observed among experimental groups at W1 (Figure 5b).

Regarding the differences among the animals with different devices and protocols after surgery, a significant overall decrease in body weight was observed from the first postoperative week onwards, regardless of the implanted device (Figure 5a,b). However, in the particular case of W3 the decrease was only found in APP animals implanted with the miniaturized and Alzet pumps compared to naïve APP (*p* APP naïve vs. miniaturized = 0.001; *p* APP naïve vs. Alzet = 0.013; Figure 5b). Finally, at W8 no differences were detected between animals implanted with the miniaturized device compared to naïve, regardless of the genotype (*p* > 0.05 for all comparisons; Figure 5b), indicating a tendency for body weight to normalize over time.

Taking the evolution of body weight for each experimental group individually (Figure 5c), we can notice a striking drop at W3 compared to W1 in the animals implanted with the original device (*p* WT = 0.018; *p* APP = 0.018; Figure 5c), a moderate decrease in those with the miniaturized device (*p* WT = 0.097; *p* APP < 0.001; Figure 5c), and no differences in those with the Alzet pump (*p* WT = 0.202; *p* APP = 0.086; Figure 5c). Interestingly, animals implanted with the miniaturized device regained basal body weight at W8 (*p* = 0.480; Figure 5c), results that are consistent with those described above.

To better understand and visualize the impact of the different implantable devices and protocols employed on animal body weight, we also included the analysis of the specific percentage change at different timepoints compared to W1 or basal body weight (Figure 6), where the significant loss in body weight at W3 is even more prominent than in the previous analysis and representations (*p* WT W0 = 0.035; *p* WT W3 = 0.002; *p* APP W0 = 0.049; *p* APP W3 < 0.001; Figure 6). In this regard, and in agreement with the aforementioned results, the percentage change in body weight decreased in all animals undergoing stereotaxic surgery, regardless of the implanted device at W3 (*p* WT naïve vs. original= 0.001; *p* WT naïve vs. miniaturized= 0.004; *p* WT original vs. Alzet = 0.045; *p* WT miniaturized vs. Alzet = 0.044; *p* APP naïve vs. original < 0.001; *p* APP naïve vs. miniaturized = 0.002; *p* APP original vs. Alzet = 0.006; Figure 6). Furthermore, this plot evidenced the remarkable drop in APP mice implanted with the original device, which largely contributed to us considering and applying a humane endpoint in these animals and terminating the experiment (Figure 6).

### 3.3. Optimized Protocol and Implantation of Smaller Devices Significantly Improve Animal Welfare and Enable Longer and Safer Experimental Periods

Animal welfare was monitored intensively after surgery and recorded weekly during the remainder of the experiment, following a customized scoresheet for intrathecal device implantation (Table 4, Figure 7). Among others, the welfare monitoring sheet included parameters of general condition, physical appearance and posture, nutritional and hydration status, spontaneous behavior, and surgery-related indicators (see details in Table 4).

Regarding the overall welfare assessment scores of each experimental group (Figure 7a), higher scores were observed in animals undergoing stereotaxic surgery in the weeks following surgery compared to naïve animals, reaching the highest values at postoperative W3 (Figure 3a). These values were markedly elevated in both WT and APP mice implanted with the original device and following the traditional protocols (Figure 7a). In fact, significant statistical differences were detected among animals implanted with the original device with respect to the rest of the experimental groups at W3 (*p* WT naïve vs. original < 0.001; *p* WT naïve vs. Alzet = 0.006; *p* WT original vs. miniaturized = 0.038; *p* WT original vs. Alzet = 0.017; *p* APP naïve vs. original < 0.001; *p* APP naïve vs. miniaturized = 0.002; *p* APP original vs. Alzet = 0.006; Figure 7b). These evaluations reached values above 10 points in some animals, representing moderate to severe suffering and pain and indicating a humane endpoint and termination of the experiment at this timepoint. For animals implanted with the miniaturized device, only significant statistical differences were detected compared to the original device at W3 (*p* WT original vs. miniaturized = 0.038; *p* APP original vs. miniaturized = 0.004; Figure 7b), but significant differences were found at W8 compared to naïve mice (*p* < 0.001 for all comparisons, Figure 7b). In this case, the values were below three points, indicating the presence of some discomfort and the need for increased vigilance and compensatory measures, but without compromising animal welfare. Finally, for animals implanted with Alzet pumps, differences in welfare scores were detected at W3 compared to naïve in both genotypes (*p* WT naïve vs. Alzet = 0.006; *p* APP naïve vs. Alzet < 0.001; Figure 7b) and compared to the original device only in WT mice (*p* WT =0.017; Figure 7b). This experimental group shared similar welfare values to those obtained in mice implanted with the miniaturized device (*p* > 0.05 for all comparisons; Figure 7b), however, termination of the experiment was conducted at W4 due to the experimental design of this particular project, but not because of humane endpoint considerations.

As described above, when the pre- and post-score values of welfare assessment were plotted and compared individually for each experimental group, an increase was detected in all the experimental groups at W3 and W8 compared to baseline, including both WT and APP naïve mice (*p* WT naïve = 0.003; *p* APP naïve = 0.010; *p* WT original = 0.017; *p* APP original = 0.016; *p* APP miniaturized = 0.001; *p* WT Alzet = 0.007; *p* APP Alzet = 0.006; Figure 7c). In the latter, this increase was mainly driven by variations in the body weight of the animals, indicating that these fluctuations are a normal situation even in healthy mice (Figure 7c). Finally, it is noteworthy that no differences in animal welfare were detected in WT animals implanted with the miniaturized device over time (*p* = 0.097; Figure 7c). In addition, a trend towards improved welfare was observed in some of their APP counterparts implanted with the miniaturized device over time, although no statistical differences were found between W3 and W8 (*p* = 0.157; Figure 7c).

### 3.4. Optimized Intrathecal Implantation Procedures Do Not Appear to Negatively Affect General and Anxiety-like Behaviors for at Least Two Months after Surgery

To minimize any additional stress on animals undergoing surgery that negatively influenced their well-being, behavioral testing was conducted exclusively in the experimental groups with the best welfare status (Figure 8a,b), i.e., naïve, animals implanted with the miniaturized device and animals implanted with Alzet pumps, at W1, W4/5, and W8 (depending on the experimental design of each group), as previously described (see corresponding Material and Methods section). The elevated zero maze was used to analyze general and anxiety-like behavior (Figure 8c), assessing the total distance traveled and the percentage of time spent in the open and closed areas of the maze [41].

Regarding total distance traveled (Figure 8d), no statistical differences were detected between genotypes (i.e., WT vs. APP) in most of the experimental groups (*p* > 0.05 for all comparisons; *p* Alzet = 0.029; Figure 8d). In addition, a longer distance traveled was only evidenced in APP mice implanted with the Alzet pump compared to those implanted with the miniaturized device or naïve ones at W1 and W4/5 (W1: *p* naïve vs. Alzet = 0.001, *p* miniaturized vs. Alzet = 0.004; W4/5: *p* naïve vs. Alzet = 0.006, *p* miniaturized vs. Alzet = 0.004; Figure 8d), whereas no differences were found in the rest of the comparisons at any of the timepoints analyzed (*p* > 0.05 for the remainder comparisons; Figure 8d). These results clearly indicate that optimized intrathecal implantation does not limit or reduce animal movement.

In terms of percentage of time spent in open and closed areas, most of the experimental groups had a strong preference for closed areas within maze corridors except WT mice implanted with the miniaturized device (*p* WT naïve W1 = 0.008, *p* WT naïve W4/5 = 0.015, *p* WT naïve W8 = 0.028, *p* APP naïve W1 = 0.013, *p* APP naïve W2 = 0.005, *p* APP naïve W8 = 0.042, *p* WT miniaturized W1, W4/5 and W8 = 0.109, *p* APP miniaturized W1 = 0.005, *p* APP miniaturized W4/5 = 0.008, *p* APP miniaturized W8 = 0.011, *p* WT Alzet W1 = 0.005, *p* WT Alzet W4/5 = 0.005, *p* APP Alzet W1 = 0.005 and *p* APP Alzet W4/5 = 0.005; Figure 8e and Appendix A). This result aligns with normal unprotected and elevated space avoidance behavior reported in mice [41]. Regarding the analysis of novel environment exploration, inferred as the percentage of time spent in open areas (Figure 8d), differences were only found between APP mice implanted with Alzet pumps compared to their counterparts implanted with the original device or the naïve ones at W1 (*p* naïve vs. Alzet = 0.007; *p* miniaturized vs. Alzet= 0.005; Figure 8d), with the percentage being always lower in the Alzet-implanted mice (Figure 8d). In addition, WT animals implanted with the miniaturized device also showed a lower percentage of time spent in open areas at W8 compared with WT naïve (*p* = 0.024; Figure 8d). These findings could indicate a possible unfavorable effect of the intrathecal implantation surgery on the particular behavior of novel environment exploration, although further investigation is needed (Figure 8d). Nevertheless, the overall results demonstrated that the refined intrathecal implantation procedures do not adversely influence the neurobehavioral functions analyzed in the implanted animals, even after two months of implantation.

### 3.5. Validation of Cannula Placement Is Required after Experiment Termination to Confirm Correct Implantation at the Stereotaxic Coordinates of Interest

To verify the correct placement of the intrathecal cannula at the specific AP, ML, and DV stereotaxic coordinates, two different approaches were performed. First, in vivo infusion and visualization of blue dextran dye in the mouse cisterna magna (Figure 9a,b) demonstrated both the correct positioning of the intracerebroventricular cannula in the mouse right ventricle and the flow of the dye solution through the device system and cannula into the ventricular system (Figure 9). In addition, post mortem dark blue visualization of blue dextran within the entire anatomy of the mouse ventricular system also verified the correct intrathecal implantation and infusion of the cannula (Figure 9c–j). Finally, histological analysis with Toluidine blue staining in coronal brain sections also confirmed the correct placement of the intracerebroventricular cannula in the stereotaxic coordinates of interest, compared to and with reference to the most widely employed mouse brain atlases: Paxinos and Franklin [37] and Allen Brain Mouse Atlas (Figure 10).

## 4. Discussion

Over the past decades, significant advances have been made in stereotaxic neurosurgery in rodents, driven by the need to adhere to higher ethical standards. This progress requires not only a high level of knowledge and technical skills, but also deep understanding and awareness of ethics, animal welfare, and the importance of minimizing the number of animals used to obtain robust conclusions, as outlined in the 3Rs principles of “refinement” and “reduction” [13].

While certain general recommendations can be universally applied to all stereotaxic surgeries, such as pain management during and after surgery, correct determination of stereotaxic coordinates, and using appropriate aseptic techniques [1], other considerations must be tailored on a case-by-case basis. In the context of our study, which involves the long-term implantation of intracerebroventricular or intrathecal cannula in small rodents, two critical steps are pivotal for successful experimentation. The first step involves securely fixing and stabilizing the cannula on the surface of skull. Previous existing protocols described several methods employing dental cement, screws, and cyanoacrylate (Refs. [1,9,10], among others). However, based on our experience and the literature, these fixation methods have proven to be less satisfactory due to the mismatch between the relatively large and flat surface of the cannula and the round-shaped skull surface of mice [10]. To address this issue, we proposed a refined stereotaxic surgery protocol that uses a UV light-curing resin that can be molded and individually adapted to each mouse in combination with cyanoacrylate adhesive tissue. The use of this resin offered several advantages, including shorter surgical intervention times due to the rapid curing time, lower incidence of surgical problems such as skin necrosis or infection (reported with the use of dental cement), and stable and secure attachment of the cannula to the skull for prolonged periods. Importantly, these improvements contribute to minimizing humane endpoint applications to very low levels, in line with the “reduction” principle of the 3Rs [13]. Additionally, our findings suggest that modifying the dimensions of the implantable devices may also be contributing to improve surgical outcomes. Therefore, careful consideration of device dimensions is crucial before designing and performing specific surgeries, especially in mice, where size constraints may arise. Importantly, a limitation of our study is that we cannot attribute the observed improvements in long-term implantation exclusively to the refined stereotaxic protocol, since both modifications (surgical refinement and changes in device dimensions) were implemented simultaneously. However, in view of the fact that with the optimized protocol we successfully overcame the most significant and critical surgical complication reported [2], which involved cannula detachment triggering the application of humane endpoint, we strongly believe that protocol refinement is the key factor mostly influencing the observed improvements. Nevertheless, additional research would clarify the level of contribution of each modification.

The second crucial step for the success of long-term intrathecal implantation studies involves careful and thorough assessment and recording of animal welfare during the immediate and long-term postoperative periods. To our knowledge, no studies have proposed or described a specific protocol to accurately monitor the animal welfare of mice undergoing such stereotaxic surgeries. Previous works have relied on individual parameters such as changes in body weight, physical appearance, or behavior related to the specific brain structure in which the cannula was implanted (Refs. [39,40], among others). In our study, we designed a standardized scoresheet for effective and accurate follow-up of the animal’s overall well-being, covering a wide range of indicators related to general condition, physical appearance and posture, nutritional and hydration status, spontaneous behavior, and the surgical procedure itself [15,16,17,18,19,20,21,22,23]. Our results corroborate the fact that relying on a single parameter is insufficient in most cases. Conversely, the use of diverse indicators provides a holistic view of the animal’s actual well-being [16,17,43]. Additionally, close monitoring has enabled us to promptly apply compensatory measures whenever necessary, thereby alleviating any discomfort or pain experienced by the animals in a timely manner and allowing studies to be extended for a longer period of time.

Although the proportion of drop-out animals due to the application of humane endpoint has reached a remarkably low level (below 5% of cases) in our studies, and we have significantly enhanced the welfare of animals undergoing long-term intrathecal implantation, we remain committed to refining our procedures further with the aim of improving animal welfare and increasing reproducibility. It is important to emphasize that maximizing animal welfare in research is crucial not only from ethical and legal perspectives, but also because it positively impacts the quality of scientific outcomes derived from these animals, thus benefiting the scientific community.

## 5. Conclusions

In summary, our study provides a refined protocol for safe intracerebroventricular cannula fixation over extended experimental periods, as well as a customized protocol to accurately monitor animal welfare in mice undergoing long-term intrathecal device implantation, among other notable findings. The inclusion of these refined steps helps to improve the safety and reproducibility of long-term experiments involving intracerebroventricular cannula fixation, providing a valuable contribution to the neuroscience community and research.

## Figures and Tables

**Figure 1 animals-13-02627-f001:**
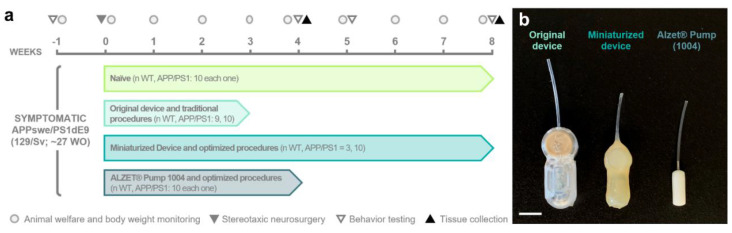
Experimental design and devices used in this study. (**a**) Schematic timeline of the experimental procedures and groups included in the study. (**b**) Photograph of the different implantable devices employed (from left to right): original device, miniaturized device, and a commercial ALZET^®^ Micro-Osmotic Pump 1004 (Cupertino, CA, USA). Note the difference in dimensions among the implantable devices. The original and miniaturized devices are prototypes of a medical device designed to selectively filter cerebrospinal fluid. Scale bar: 1 cm. WO, weeks old; WT, wild-type mice.

**Figure 2 animals-13-02627-f002:**
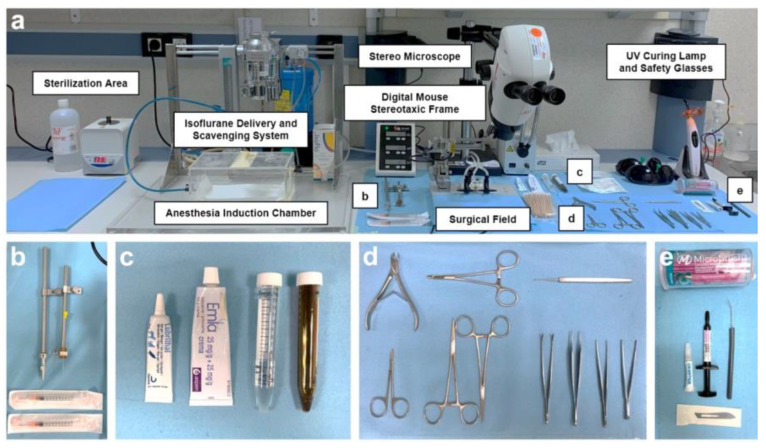
Operating room setup and surgical instrument arrangement. (**a**) Overview of the operating room showing the main equipment and instruments required. (**b**–**e**) Close-up views highlighting specific instruments and components used during surgery: cannula and stereotaxic needle holders (**b**); ocular and topical anesthetic ointments, 70% (*v*/*v*) ethanol, and 1% (*v*/*v*) povidone-iodine solution (**c**); surgical instruments (**d**); and cyanoacrylate tissue adhesive, UV light-curing resin, and applicators (**e**).

**Figure 3 animals-13-02627-f003:**
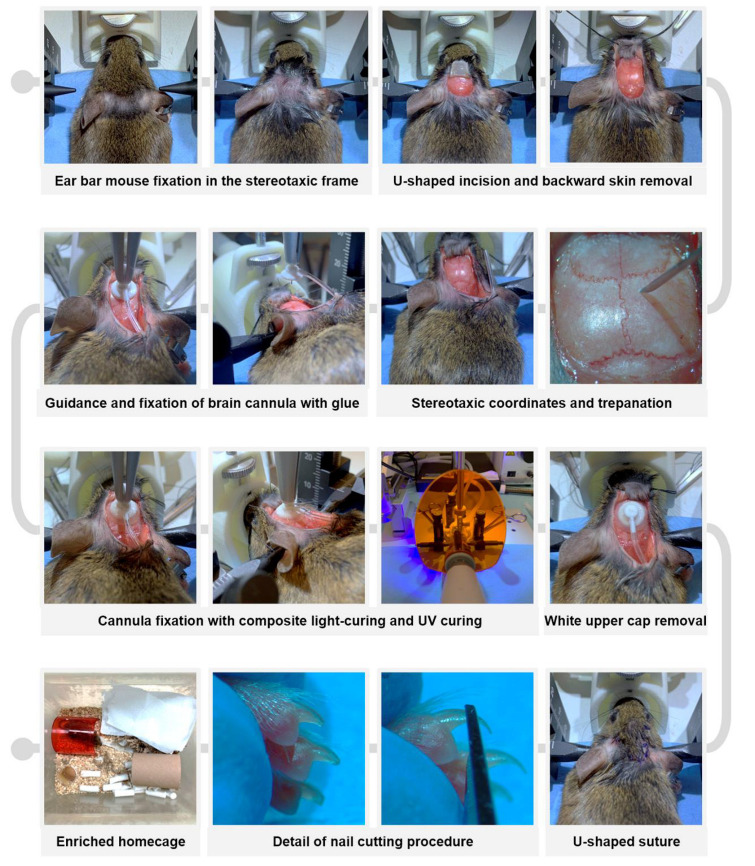
Roadmap of key steps in the refined neurosurgical protocol for long-term implantation of intrathecal cannula in mice.

**Figure 4 animals-13-02627-f004:**
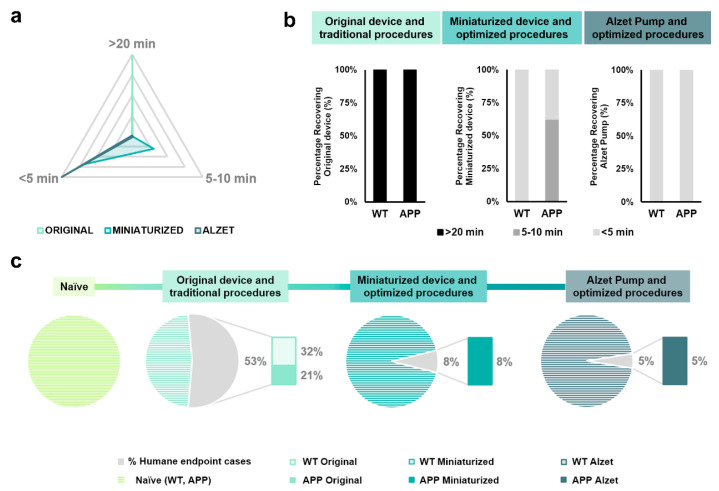
Qualitative analyses of the effect of changes in implantable device dimensions and stereotaxic protocol followed on both the recovery time after surgery and the percentage of humane endpoint application. (**a**) Prism representation of the percentage of animals in each range of recovery time (i.e., >20 min, 5–10 min and <5 min, individually plotted in each vertex) for each experimental group, regardless of genotype. Each inner grey prism line represents 25%. (**b**) Quantification of the percentage of animals in each range of recovery time separated both by implanted device and protocol, and by genotype. Note that 100% of the animals implanted with the original device and Alzet pump required more than 20 min and less than 5 min to recover from surgery, respectively; whereas all animals implanted with the miniaturized device needed less than 10 min. (**c**) Quantification of the percentage of humane endpoint application for each experimental group. Note that changes in the implanted device and refinement of the protocol significantly reduced the application of humane endpoint from 53% in animals implanted with the original device to 5% in those implanted with Alzet pumps. Specific sample sizes and Fisher exact test *p*-values are provided in Table 1 and Table 5, respectively.

**Figure 5 animals-13-02627-f005:**
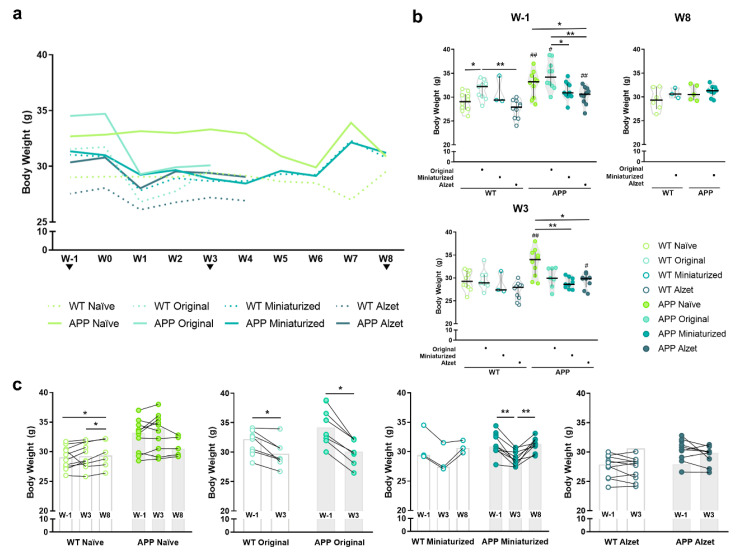
Effect of changes in the dimensions of implantable device and the refined stereotaxic protocol on the body weight at different timepoints. (**a**) Representation of the median body weight throughout the duration of the experiment of the different experimental groups included in this work (based on the implanted device and genotype). Note the striking reduction in body weight in all the animals undergoing stereotaxic surgery from the first postoperative week, particularly in those implanted with the original device; and the partial recovery in mice implanted with the miniaturized device from postoperative W6 onwards. No statistical test has been performed for these data over time due to the incompatibility with the wide variety of experiment terminations. (**b**) Quantification of animal body weight at timepoints shared by most experimental groups, i.e., W1, W3, and W8. (**c**) Pre- and post-representations of body weight at W1 versus (vs.) W3 vs. W8 or W1 vs. W3, for each experimental group. Note that animals implanted with the miniaturized device regain the body weight loss observed at W3 over time. Specific sample size is provided in Table 1. Data expressed as median and IQR. Kruskal–Wallis test for multiple comparisons (W1 and W3 in (**b**)); Mann–Whitney U test for independent pairwise comparisons (WT vs. APP comparisons in b; W8 in (**b**)); Friedman test for repeated-measures multiple comparisons (naïve and miniaturized in **c**); and Wilcoxon test for dependent pairwise comparisons (original and Alzet in (**c**)). * *p* < 0.05; ** *p* < 0.01, for differences between experimental groups based on the implanted device and multiple comparisons; # *p* < 0.05; ## < 0.01, for differences between WT and APP genotypes in b (see Appendix A for detailed analysis). W, week.

**Figure 6 animals-13-02627-f006:**
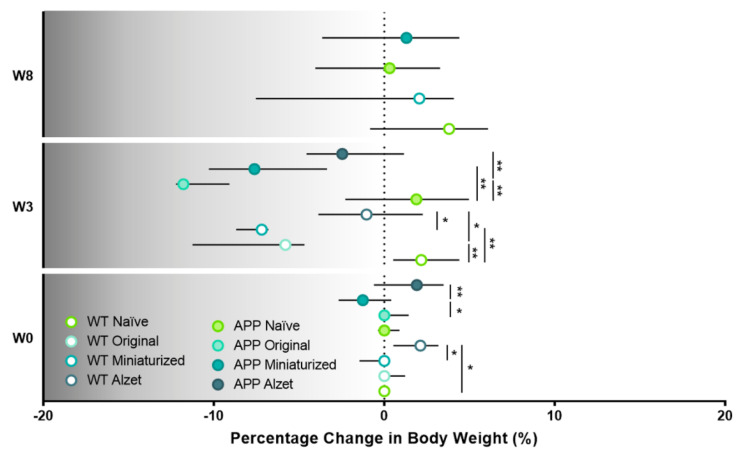
Effect of the different implantable devices and the refined stereotaxic protocol on the percentage change in animal body weight at different timepoints. Note the striking drop in APP mice implanted with the original device at W3 leading to us considering and applying a humane endpoint in this experimental group. Specific sample size is provided in Table 1. Data expressed as median and IQR. Kruskall–Wallis test for multiple comparisons (W0 and W3) and Mann–Whitney U test for independent pairwise comparisons (W8). * *p* < 0.05; ** *p* < 0.01. W, week.

**Figure 7 animals-13-02627-f007:**
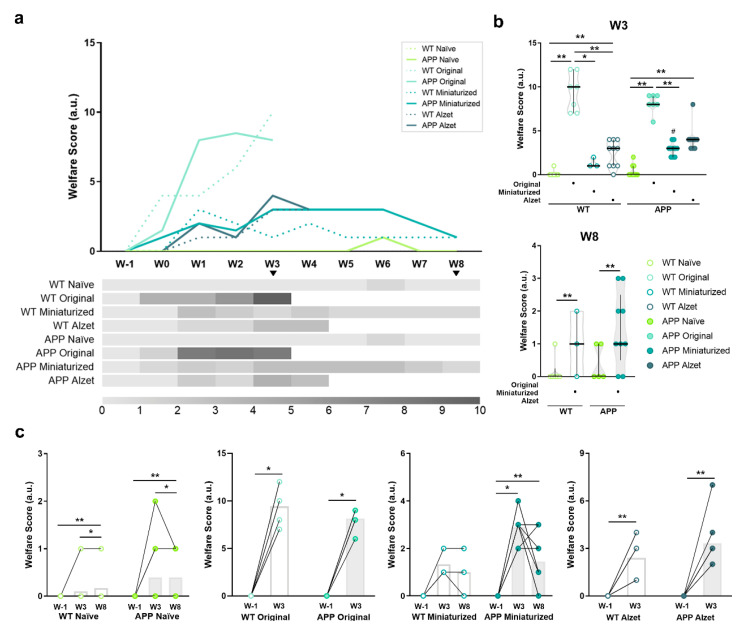
Effect of changes in the dimensions of implanted device and the refined stereotaxic protocol on animal welfare throughout the experiment. (**a**) Representation (top) and heatmap (bottom) of the median welfare assessment scores throughout the duration of the experiment for each experimental group. The higher the scores, the more compromised the welfare of animals. It is important to point out the elevated scores obtained in both WT and APP mice implanted with the original device and following the traditional protocol at W2 and W3, which resulted in the application of humane endpoint and termination of the experiment for this experimental group at this timepoint. No statistical test has been performed for these data over time due to the incompatibility with the wide variety of experiment termination. (**b**) Quantification of animal welfare at timepoints shared by most experimental groups, i.e., W3 and W8. (**c**) Pre- and post-representations of animal welfare scores at W1 vs. W3 vs. W8 or W1 vs. W3, for each experimental group individually. Specific sample size is provided in Table 1. Data expressed as median and IQR. Kruskal–Wallis test for multiple comparisons (W3 in (**b**)); Mann–Whitney U test for independent pairwise comparisons (WT vs. APP comparisons in b; W8 in (**b**)); Friedman test for multiple comparisons with repeated measures (naïve and miniaturized in (**c**)); and Wilcoxon test for dependent pairwise comparisons (original and Alzet in (**c**)). * *p* < 0.05; ** *p* < 0.01, for differences between experimental groups based on the implanted device and multiple comparisons; # *p* < 0.05; for differences between WT and APP genotypes in (**b**) (see Appendix A for detailed analysis).

**Figure 8 animals-13-02627-f008:**
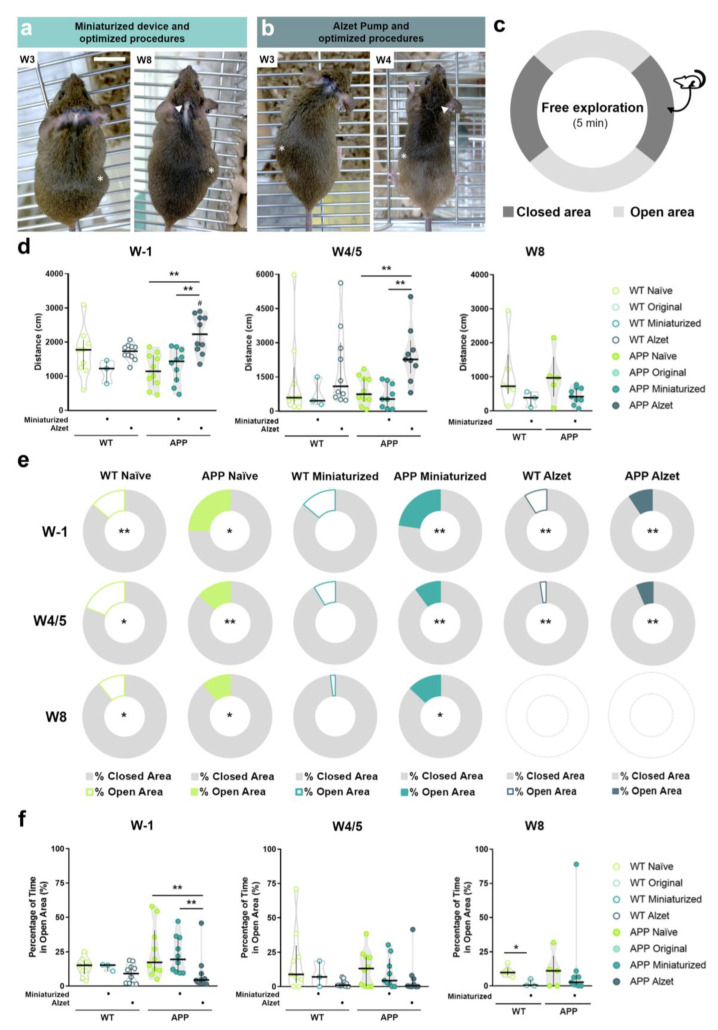
Effect of implantation of smaller devices following refined intrathecal implantation procedures on general and anxiety-like behaviors. (**a**,**b**) Dorsal views of two different mice implanted with the miniaturized device at W3 and W8 (**a**), and with the Alzet pump at W3 and W4 (**b**), following in both cases the optimized surgery protocol. Note the good general appearance of the animals and of the healed wound, especially at the end of the experiment where it is almost unnoticeable (arrowheaded). The asterisks point out the subcutaneous pocket where device is implanted. (**c**) Schematic representation of the elevated zero maze apparatus. (**d**) Quantification and comparisons of the total distance traveled in the maze at W1, W4/5 and W8. (**e**) Quantification and ring plot of the median percentage of time spent in closed vs. open areas individually for each experimental group and timepoint (see Appendix A for detailed analysis). (**f**) Quantification and comparisons of the percentage of time spent in open area among the different experimental groups at W1, W4/5 and W8. Specific sample size is provided in Table 1. Data expressed as median and IQR. Kruskal–Wallis test for multiple comparisons (W1, W4/5 in (**d**) and (**f**)); Mann–Whitney U test or Wilcoxon test for independent (W8 in d and W8 in (**f**)) and dependent (% of open vs. % of closed in (**e**)) pairwise comparisons; respectively. * *p* < 0.05; ** *p* < 0.01, for differences between experimental groups based on the implanted device and multiple comparisons; # *p* < 0.05, for differences between WT and APP genotypes in (**d**).

**Figure 9 animals-13-02627-f009:**
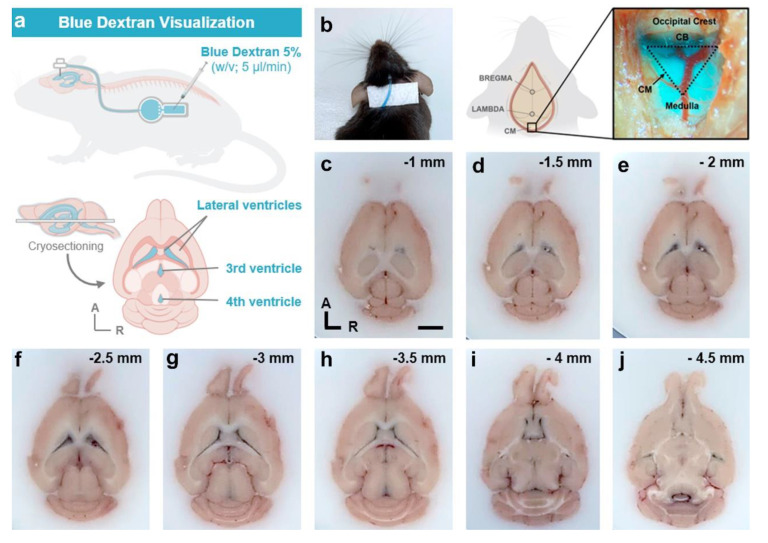
Verification of intracerebroventricular cannula implantation by infusion and visualization of blue dextran. (**a**) Experimental design of blue dextran (5%, *v*/*v* in aCSF) study. (**b**) Dorsal view of in vivo visualization of blue dextran in the connecting tube between the cannula and the device (left) and in the mouse cisterna magna (right). (**c**–**j**) Transversal cryosections (spaced 0.5 mm apart) of mouse brain infused with blue dextran. Note that blue color of dextran is visualized throughout the anatomy of the mouse ventricular system, indicating correct placement and flow of the cannula. Scale bar (**c**–**j**): 2 mm. A, anterior; CB, cerebellum; CM, cisterna magna; R, right.

**Figure 10 animals-13-02627-f010:**
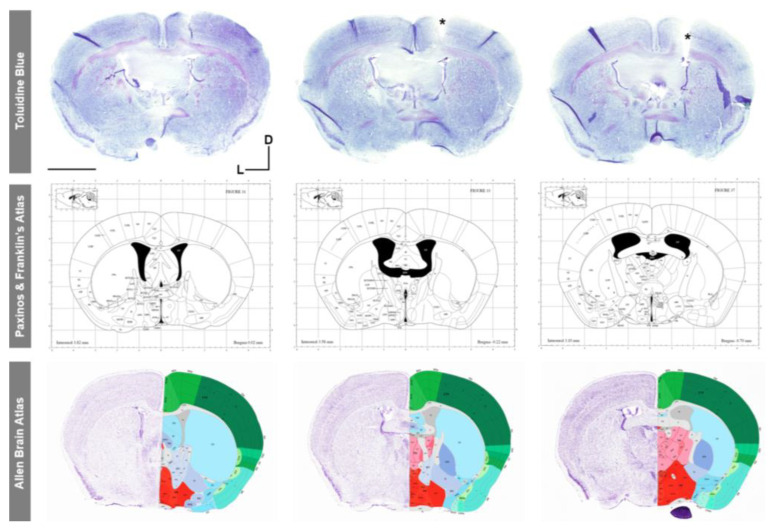
Histological validation of the specific stereotaxic coordinates (AP = −0.70 mm; ML = −1.26 mm; and DV = −2.5 mm) used for intracerebroventricular cannula implantation with Toluidine blue staining and reference to mouse brain atlases. Images of middle and bottom rows obtained from [37] and http://mouse.brain-map.org/ (accessed on 1 July 2023), respectively. Scale bar: 2 mm. D, dorsal; L, left; *, cannula placement.

**Table 1 animals-13-02627-t001:** Specific sample sizes of each experimental group for different parameters at different timepoints.

Parameter and Time Analyzed	Naïve		Original Device		Miniaturized Device		Alzet Pump ^a^	
	WT	APP	WT	APP	WT	APP	WT	APP
Recovery time after surgery	10	10	9	10	3	10	10	10
Humane endpoint application	10	10	9	10	3	10	10	10
Body weight at W1	10	10	9	10	3	10	10	10
Body weight at W3	10	10	7	7	3	9	10	9
Body weight at W8	5	5	-	-	3	9	-	-
% Change in body weight at W0	10	10	9	10	3	10	10	10
% Change in body weight at W3	10	10	7	7	3	9	10	9
% Change in body weight at W8	5	5	-	-	3	9	-	-
Welfare assessment score at W3	10	10	7	7	3	9	10	9
Welfare assessment score at W8	5	5	-	-	3	9	-	-
Behavioral testing at W1	10	10	-	-	3	9	9	10
Behavioral testing at W4/5	10	10	-	-	3	9	9	9
Behavioral testing at W8	6	5	-	-	3	9	-	-

Legend: W, week. ^a^ “Alzet pump” refers to micro-osmotic pump model 1004 (ALZET^®^, Cupertino, CA, USA).

**Table 2 animals-13-02627-t002:** Comparison of design parameters among devices and the commercial micro-osmotic pump used in this study.

Parameter	Original Device	Miniaturized Device	ALZET^®^ Pump 1004 ^a^
Weight (filled; g)	4.29 ± 0.41	2.25 ± 0.15	0.56 ± 0.06
% Weight (with respect to animal body weight ^b^; %)	14.16 ± 1.35	6.93 ± 0.46	1.85 ± 0.19
Total Volume (cm^3^)	5.92	2.54	0.54
Maximum L (cm)	3.5	2.9	1.5
Maximum W (cm)	1.5	1.3	0.6
Maximum H (cm)	1.5	0.9	0.6
Coated Material	Polyetheretherketone (PEEK; ISO 13485 [29])	LOCTITE^®^ SI 5248 silicone (Henkel, Düsseldorf, Germany; ISO 10993 [30])	Cellulose ester blend
Catheter Tube Length and Outside Diameter (cm)	3 ± 0.2, 0.11 ± 0.008	3 ± 0.2, 0.11 ± 0.008	3 ± 0.2, 0.11 ± 0.008
Catheter Tube Material and Configuration	Polyvinyl chloride (MG), pre-attached	Polyvinyl chloride (MG), pre-attached	Polyvinyl chloride (MG), attachable
Brain Infusion III Cap Cannula Dimensions (L, W, H; cm) ^c^	0.59, 0.59, 0.2	0.59, 0.59, 0.2	0.59, 0.59, 0.2
Brain Infusion III Cap Material ^c^	Polycarbonate	Polycarbonate	Polycarbonate
Brain Infusion III Cannula Dimensions (L, W, H; cm) ^c^	0.3, 0.031, 0.031	0.3, 0.031, 0.031	0.3, 0.031, 0.031
Brain Infusion III Cannula Material ^c^	Stainless steel	Stainless steel	Stainless steel

Legend: H, height; L, length; MG, medical grade; W, width; ISO, International Organization for Standardization. ^a^ Data obtained from Micro-osmotic Pump Model 1004 (ALZET^®^, Cupertino, CA, USA) Datasheet; ^b^ Mean ± SEM Body weight of 7-months old male mouse (30.30 ± 0.16); ^c^ Data obtained from Brain Infusion Kit III 1–3 mm (ALZET^®^) Datasheet.

**Table 3 animals-13-02627-t003:** Detailed dimensions of the different device components.

Parameter	Original Device	Miniaturized Device	ALZET^®^ Pump 1004 ^a^
Apheresis Module Dimensions (L, W, H; cm)	1.4, 1.5, 1.5	1.3, 1.3, 0.9	-
Apheresis Module External Volume (cm^3^)	3.150	1.521	-
Apheresis Module Internal Volume (µL)	235.5	235.5	-
Reservoir Dimensions (L, W, H; cm)	2.1, 1.2, 1.1	1.6, 0.8, 0.8	1.5, 0.6, 0.6
Reservoir External Volume (cm^3^)	2.772	1.024	0.540
Reservoir Internal Volume (µL)	100	100	100
Total Volume (cm^3^)	5.922	2.545	0.540

Legend: H, height; L, length; W, width. ^a^ Data obtained from micro-osmotic pump model 1004 (ALZET^®^, Cupertino, CA, USA) datasheet.

**Table 4 animals-13-02627-t004:** Scoresheet for animal welfare monitoring criteria in mice undergoing intrathecal catheter surgery.

Parameters		Score
**General Condition, Physical Appearance, and Posture**		
Smooth and shiny fur, clean forelimbs, and nose		**0**
Presence of piloerection, unkempt fur		**1**
Abnormal posture (abdominal curvature and/or kyphosis, increased muscle tone)		**2**
Skin lesions unrelated to surgery (ear dermatitis, scratches, excessive barbering)		**2**
**Nutritional and Hydration Status**		
Unaffected or increased		**0**
Dehydration signs	<2 s after pinching the back skin	**1**
	>2 s after pinching the back skin	**2**
Animal Body weight	1 ≤ 10% weight loss	**1**
	10–20% weight loss	**4**
	>20% weight loss (duration >4 postoperative days)	**8**
**Spontaneous Behavior**		
Normal behavior (sleeping, exploration, grooming, nesting, interaction with environmental enrichment objects)		**0**
No use of enrichment objects, no nesting behavior		**1**
Impairment of motor function (hypo-locomotion)		**4**
Lethargy and/or slight loss of balance		**8**
Hind-limbs paralysis, tremors and/or signs of vestibulocochlear dysfunction		**12**
**Surgery-specific Parameters**		
Clean and dry surgical incision, no signs of infection, no pain or signs of distress		**0**
Scratches around the scar or slight redness		**1**
Redness and/or necrosis of the skin around the scar		**3**
Grimace scale	Moderate	**4**
	Severe	**6**
Suture opening Grade I (loose stitches with closed and healed wound)		**3**
Suture opening Grade II (loose stitches with unhealed wound)		**6**
Suture opening Grade III (open unhealed wound)		**10**
Suture opening Grade IV (*Brain Infusion Kit cannula* disconnected)		**12**

Score 0: normal physical condition, no pain. Scores 1–6: presence of discomfort indicating need for sustained observation and increased vigilance, measures such as providing moist and palatable food, administering analgesia, antibiotics, and subcutaneous saline or glucose may be necessary. Scores 6–10: significant suffering and pain indicating need for compensatory measures, subcutaneous glucose, seeking veterinary advice, re-operating scar with recovery in intensive care unit, and considering euthanasia if maintained for four consecutive days. Scores 10–12: moderate to severe suffering and pain, consider a humane endpoint if compensatory measures do not resolve complications and scores are maintained for 24 h. Scores > 12: severe suffering indicative of a humane endpoint application and termination of the experiment.

**Table 5 animals-13-02627-t005:** Comparisons and specific *p*-values for statistical qualitative analyses of recovery time and humane endpoint application parameters.

Parameter	Comparisons and *p*-Values			
Recovery time	Original Device		Miniaturized Device	Alzet Pump
>20 min	**<0.001**		**0.002**	**<0.001**
5–10 min	0.284		**0.003**	0.151
<5 min	**<0.001**		0.341	**<0.001**
	Naïve	Original Device	Miniaturized Device	Alzet Pump
Humane endpoint	**0.029**	**<0.001**	0.681	0.159

Bold values indicate statistical significance at the *p* < 0.05 in Fisher exact test, i.e., compared variables are related.

## Data Availability

The data that support the findings of this study are available from the corresponding authors upon reasonable request.

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
