# Peer review of "Refining Stereotaxic Neurosurgery Techniques and Welfare Assessment for Long-Term Intracerebroventricular Device Implantation in Rodents"

_animals, 2023, doi:10.3390/ani13162627_

Round 1

Reviewer 1 Report

Comments:

In this work the Authors accurately describe a refined protocol for safe intracerebroventricular cannula fixation over extended experimental periods, as well as customized protocol to monitoring animal welfare in mice undergoing long-term intrathecal device implantation. The originality of this study is the assessment of a scoresheet to accurately monitor animal well-being during long-term implantation.

However, some revision of the study is needed:

Introduction:

Line 62. Authors should better explain why the size of the device affects the mortality rate;

Line 97. “Firstly, we have modified the dimensions of the implantable devices we have previously used, significantly reducing the ratio between devices weight and animal body weight, while ensuring the same functionality.” In this case Authors should better explain this concept.

Materials and method:

Line 115. In this experimental protocol Authors used male mice. You can know if there are differences between male and female as to welfare post- surgery?

Line 208. Why it has been injected sterile saline solution intraperitoneally prior the surgery? 

Line 212. It is not clear why the isolation of animal after surgery decrease the distress due to aggression and potential wound opening/infection.

Line 253. What is the “pre-warming

Finally, in material and methods the aseptic procedures prior to surgery should better clarified.

 Result

The relationship between the size of the device and the decrease of the body weight can be better explained.

Finally, some references are dated. More recent study should be added to the text.

Reviewer 2 Report

In this manuscript, the authors present an optimized method for the safe intracerebroventricular or intrathecal long-term device implantation, with a strong emphasis on animal welfare. The paper introduces a compelling and contemporary approach that explores the potential benefits of reducing animal usage, enhancing experimental data quality, and improving reproducibility, all of which contribute to a more ethical use of animals in research. The clarity of the manuscript and the detailed bullet point description provided in the supplementary material demonstrate the authors' careful consideration and dedication to a robust procedure.

As a reviewer, I find the manuscript highly valuable and necessary, particularly given the prevalence of experiments utilizing such procedures. My suggestion is that the authors endeavor to keep the manuscript updated regularly, as they appear to be actively engaged in ongoing improvements (lines 762-763). A continuously updated version will ensure that the latest developments and refinements are captured, benefiting the scientific community.

Furthermore, I propose that the authors consider creating and sharing a comprehensive video detailing all the procedures. Such a video would undoubtedly prove to be an invaluable resource for neuroscientists worldwide, providing them with visual guidance and a better understanding of the methodology. This step would further enhance the dissemination and adoption of their optimized approach, thereby advancing the field and promoting ethical research practices.

Author Response

The authors would like to thank the reviewer for her/his kind words regarding the manuscript. We will try to keep any improvements we may introduce to this procedure up to date, as suggested by the reviewer.
We also consider the possibility of producing a video to be an excellent idea, so we are available to the editors to carry it out.

Reviewer 3 Report

This paper describes the authors’ work to refine the stereotactic surgery required to implant an intracerebroventricular in mice, detailing in particular changes to the device itself and comparing this to the most common commercial solution for this type of work as well as the monitoring performed. The manuscript provides a detailed breakdown of the surgery performed and presents the welfare-related measures taken.

 A paper of this type, describing a (relatively) common surgical approach in detail with suggestions for refinements, will be an invaluable resource to the community. That the data themselves are clearly presented makes this of even greater value, so the authors are to be commended for taking the time to put this together.

Aside from a single typo (line 60 of page 2, “human” should likely be “humane”) I have no suggestions to improve this manuscript.

Author Response

The authors would like to thank the reviewer for her/his kind words regarding the manuscript and for reviewing it thoroughly.
We have corrected the typo found by the reviewer on line 60 in the new version of the manuscript.